# Ellagic Acid from Hull Blackberries: Extraction, Purification, and Potential Anticancer Activity

**DOI:** 10.3390/ijms242015228

**Published:** 2023-10-16

**Authors:** Jialuan Wang, Fengyi Zhao, Wenlong Wu, Lianfei Lyu, Weilin Li, Chunhong Zhang

**Affiliations:** 1Jiangsu Key Laboratory for the Research and Utilization of Plant Resources, Institute of Botany, Jiangsu Province and Chinese Academy of Sciences (Nanjing Botanical Garden Mem. Sun Yat-Sen), Qian Hu Hou Cun No. 1, Nanjing 210014, China; wjl163youxiang@163.com (J.W.); zhaofengyi92@163.com (F.Z.); 1964wwl@163.com (W.W.); njbglq@163.com (L.L.); 2Co-Innovation Center for Sustainable Forestry in Southern China, College of Forestry, Nanjing Forestry University, 159 Longpan Road, Nanjing 210037, China

**Keywords:** blackberry, ellagic acid, extraction, purification, anticancer activity

## Abstract

Ellagic acid (EA) is present at relatively high concentrations in many berries and has many beneficial health effects, including anticancer properties. To improve the development and utilization of blackberry fruit nutrients, we divided Hull blackberry fruits into five growth periods according to color and determined the EA content in the fruits in each period. The EA content in the green fruit stage was the highest at 5.67 mg/g FW. Single-factor tests and response surface methodology were used to optimize the extraction process, while macroporous resin adsorption and alkali dissolution, acid precipitation, and solvent recrystallization were used for purification. The highest purity of the final EA powder was 90%. The anticancer assessment results determined by MTT assay showed that EA inhibited HeLa cells with an IC_50_ of 35 μg/mL, and the apoptosis rate of the cells increased in a dose-dependent manner, with the highest rate of about 67%. We evaluated the changes in the mRNA levels of genes related to the EA-mediated inhibition of cancer cell growth and initially verified the PI3K/PTEN/AKT/mTOR pathway as the pathway by which EA inhibits HeLa cell growth. We hope to provide a theoretical basis for the deep exploration and utilization of this functional food.

## 1. Introduction

Blackberries (*Rubus* spp.) are fruits of interest owing to their high content of anthocyanins and ellagitannins as well as other phenolic compounds that contribute to their good biological activities [1], including antioxidant [2], anticancer [3], anti-inflammatory [4], antibacterial [5] and others. This fruit is composed of an aggregate of droplets 1–3 cm in diameter that change color from green to red to black as it ripens [6]. However, the majority of blackberry varieties have a sour taste when fresh and are not easy to store, which is detrimental to the development and utilization of blackberries. Consequently, studying the functional components of blackberries by optimizing the extraction and utilization process is important.

Reportedly, the highest concentrations of ellagic acid (EA) are found in fruits of plants of the genus Rubus [7]. In plants, EA is a biologically active polyphenolic compound that occurs naturally as a secondary metabolite in many plants, where it is produced mainly by hydrolyzing ellagitannins [8]. Structurally, EA is considered a dimeric gallic acid derivative because it constitutes a dilactone of hexahydroxy-diphenic acid (HHDP) (Figure 1) [9]. In recent decades, EA has attracted increasing attention due to its pronounced antioxidant [10,11], anticancer [12,13,14], anti-inflammatory [15,16], and antimutagenic properties [17,18]. Many studies have shown that EA can regulate a range of cell signaling pathways to prevent, mitigate or slow the progression of chronic diseases such as cardiovascular disease [19] and neurodegenerative diseases [20], diabetes [21], and cancer [13,14,22].

The evidence from epidemiologic and clinical studies suggests that a daily intake of 400–800 g of vegetables and fruits may prevent 20% or more of cancer cases. Studies in vitro, animal, and clinical studies have demonstrated the potential role of berry phenolic compounds in reducing cancer risk [23]. EA has a significant inhibitory effect on chemical-induced carcinogenesis and many other types of carcinogenesis, such as liver, lung, colon, breast, and cervical cancer [24,25,26,27,28,29]. Cervical cancer is the fourth leading cause of cancer death in women [27]. Long-term infection with human papillomavirus (HPV) is one of the causes of cervical cancer, and approximately 91% of cervical cancer patients are infected with high-risk HPV [30]. Studies have found that 16/18 are the two most prevalent HPV subtypes in patients with cervical cancer. Several studies have shown that the E6/E7 gene is the most abundant viral transcript in biopsies and HPV-positive cells from HPV-positive cervical cancer patients. E6/E7 plays a key role in the process of viral replication and carcinogenesis. The open reading frames of E6/E7 sequences are directly involved in regulating the growth and proliferation of cervical cancer cells and are closely related to apoptosis [31,32,33,34,35]. In this paper, apoptosis-related pathways and E6/E7 genes were selected. Changes in the mRNA levels of the related pathway genes were determined by qPCR, which initially revealed the related pathway of EA to inhibit the growth of HeLa cells. This provides part of the theoretical basis for the further development of EA in blackberry fruit.

EA has special physicochemical properties and low solubility in many solvents, so improving its extraction efficiency is significant for its utilization and research. In this study, the different growth stages of Hull were divided into five periods according to color, including green fruiting (S1), green to red (S2), red fruiting (S3), red to purple (S4), and ripening (S5). We determined the EA content as well as other antioxidants in fresh fruits at each stage. After determination and analysis, we selected the S1 period fruits for extraction and purification. After finally obtaining 90% pure EA powder, we further investigated the in vitro anticancer activity of EA and its possible pathway to inhibit cancer cells. It is hoped that this work will provide a theoretical basis for exploring the effective utilization value of blackberries and developing more functional foods.

## 2. Results

### 2.1. Determination of EA

The important morphological indicators and EA content in Hull fruits were measured at each ripening stage. Fifteen fruits were randomly selected to measure their weights and diameters, as shown in Figure 2b. The EA content showed a decreasing trend with fruit growth and development. Among the five stages, the highest EA content was found at the S1 stage at 5.67 mg/g (Figure 2c), while the lowest EA content was found at the S5 stage at 0.77 mg/g. Others were S2 with 1.54 mg/g, S3 with 0.89 mg/g, and S4 with 0.99 mg/g.

### 2.2. Measurement of Fruit Quality Indexes

The antioxidant capacity and antioxidant substances as well as the content of saccharides in the fruits of Hull at different coloring stages were determined, which showed that the antioxidant substances and antioxidant capacity declined gradually with fruit growth and development (Figure 3). Among them, vitamin C decreased and then increased during fruit development, and the content was the highest in the S1 stage, while anthocyanin, which is related to fruit color, gradually accumulated with fruit growth and development. The total antioxidant capacity of fruits in Hull was 857.90 U·mgprot^−1^ FW in S1 period, DPPH scavenging capacity was higher at 340.07 mg Trolox·g^−1^ FW, and the highest content of anthocyanin was 0.91 mg·g^−1^ FW in S5 stage.

Sugars are one of the important evaluation indexes of fruit flavor, which gradually accumulated with fruit growth and development, and reached the highest in the S5 period. The soluble sugar content increased linearly during fruit development, with the highest content of 61.78 mg/g in the S5 period, while the fructose content decreased gradually in the S1–S3 period, with the lowest content of 12.71 mg/g in the S3 period. The fructose content increased linearly in the middle and late stages of fruit development, with the highest content of 56.65 mg/g in the S5 period (Figure 4). EA synthesis pathway is the mangiferolic acid pathway, the starting material is phosphoenolpyruvic acid, EA can combine with sugar to form glycosides, which leads to a decrease in free EA content. The high content of antioxidants and the low content of sugars in the S1 period during fruit growth and development make it suitable as a sample for EA extraction.

### 2.3. Single-Factor Tests for EA Extraction

EA lyophilized powder was made from S1 fruits with the highest EA content. Four factors, including ethanol concentration, solid–liquid ratio, extraction time, and extraction temperature, were used to analyze the extraction process. The EA extracted under different conditions was determined and the optimal extraction conditions were studied. The results of the single-factor experiment showed that the optimal extraction conditions were a solid–liquid ratio of 1:20, an ethanol concentration of 40%, an extraction time of 20 min, and an extraction temperature of 80 °C.

The extraction efficiency of EA increased gradually with the increase in ethanol volume fraction and reached a peak extraction of 47.39 mg/g when the ethanol concentration reached 40%. However, the extraction efficiency decreased gradually when the ethanol concentration exceeded 40%. Therefore, the optimum concentration of ethanol is 40% (Figure 5a).

The solid–liquid ratio has a strong influence on the extraction efficiency of EA. The smaller the solid–liquid ratio is, the more incomplete the extraction is and the lower the extraction efficiency is; the larger the solid–liquid ratio is, the more unnecessary waste will be produced, which also reduces the extraction efficiency. When the solid–liquid ratio was 1:20, the EA had been fully precipitated, and the EA content was 38.33 mg/g. Finally, 1:20 was chosen as the optimal solid–liquid ratio (Figure 5b).

Ultrasound can destroy the cell wall and make the cell contents more soluble in the extraction solvent. As the extraction time increased, the extraction rate of EA gradually increased, but a too-prolonged extraction time would consume a lot of time and cause problems such as EA aging, so the best extraction time was 20 min (Figure 5c). When the ultrasonic extraction time was 20 min, the maximum amount of EA extracted was 51.75 mg/g.

EA is not sensitive to high temperatures and is not highly soluble in many solvents. The extraction rate and solubility increased with increasing extraction temperature. When the temperature reached 80 °C, the EA content also reached the highest value of 53.46 mg/g. However, prolonged extraction at high temperatures will lead to the aging of EA and reduce its activity. Hence, 80 °C was chosen as the optimal extraction temperature. The best extraction conditions derived from the single-factor test were solid–liquid ratio of 1:20, ethanol concentration of 40%, ultrasonic extraction temperature of 80 °C, and time of 20 min. Under these conditions, the content of the extracted EA lyophilized powder was 55.20 mg/g (Figure 5d).

Solvent extraction is the earliest and most classical method. And ultrasonic waves have the advantages of penetration and cavitation, which can make the liquid molecules collide and interact with each other, thus rapidly rupturing the plant cell wall and releasing phenolic compounds. Ultrasonic extraction is easier to operate, with lower instrument costs, and is more efficient than other techniques [36,37]. Because of its similar polarity to EA, acetone is difficult to react with hydrolyzed tannins compared to other solvents. Li and colleagues used acetone as an extraction solvent and determined that an 80% acetone solution (containing HCl and vitamins) with a solid–liquid ratio of 1:12 (g/mL) was the most effective for EA extraction (323 μg/g) at 80 °C and reflux extraction for 90 min [38]. Ethanol is recognized as a low-toxicity organic reagent. Wang and coworkers used anhydrous ethanol as a solvent to extract EA from raspberries by ultrasound-assisted extraction. The best extraction conditions were as follows: solid–liquid ratio of 1:14 (g/mL), ultrasonic extraction for 20 min, and extraction temperature at 80 °C, and 670.28 μg/g of EA was obtained [39].

### 2.4. Response Surface Methodology to Optimize EA Extraction

According to the results of the single-factor experiments, the best three values were selected, and the above four critical factors were optimized using the experimental design software Design-Expert 8.0.6.1 (Stat-Ease, Minneapolis, MN, USA) and the Box–Behnken design method; the related process mathematical models were established and verified. The results of EA extraction were determined according to the experimental conditions shown in Table 1 and Table 2.

By response surface analysis, the EA extraction efficiency equation in relation to the main factors was determined to be y = 55.13 − 0.37A + 1.78B − 0.84C − 0.96D + 0.076AB − 0.085AC + 0.39AD − 0.20BC − 0.71BD + 0.50CD − 1.87A2 − 3.39B2 − 3.67C2 − 2.92D2. The partial regression coefficients of the primary terms in the order of B > D > C > A indicated that the most influential factor on the EA extraction efficiency was the extraction time, followed by the solid–liquid ratio, ethanol concentration, and extraction temperature. The ANOVA results showed that the experimental model was significant with a misfit of 0.604 and an R^2^ of 0.988, which was similar to the predicted R^2^ = 0.946, indicating that the experimental results were reliable.

Using Design-Expert 8.0.6.1 software, the contour lines and response surface maps of the corresponding quadratic regression equations were obtained (Figure 6). Discretization of the slope of the response surface and the contour of the response surface can directly reflect the interaction between the factors. It can be seen from the figure that the response surfaces of extraction time and liquid–solid ratio are very steep, indicating that extraction time and solid–liquid ratio have a significant effect on the extraction volume. In addition, the contours are elliptical in shape, indicating that the interaction between them is also very significant. The best process for extracting EA was determined to be a 1:20 solid–liquid ratio, 40% ethanol concentration, 80 °C extraction temperature, and 20 min extraction time.

### 2.5. Purification of EA

The purification experiments were mainly performed by adsorption of macroporous resins, and the most suitable macroporous resin was selected by determining the static adsorption and resolution. HPD300 and HP20 had strong adsorption capacity for EA, and the adsorption capacity was stable after 5 h. The adsorption rates were about 65% and 63%, and the amounts of EA adsorbed on the resins were 13.77 mg/g and 13.43 mg/g, respectively. The results showed that the resolution of EA was completed after adding ethanol for 30 min, with the highest resolution of 93% and 88% for HP20 and HPD300, respectively (Figure 7). HP20 macroporous resin was selected for subsequent purification.

In order to separate compounds, the adsorption force of macroporous resins depends on the physical adsorption between the macroporous resins and the adsorbed substances through van der Waals forces. Macroporous resins have the advantages of high extraction rate, non-toxicity, non-harmfulness, and non-pollution of the environment. Wang’s group compared the adsorption capacity of different resins for EA, and finally chose HPD 600 for the first isolation and purification of the crude extract of EA [40]. In the experiment, the EA concentration measured at constant volume was 61%, and after the second purification by CG-161 resin, the final concentration of EA reached 80%.

The first purified powder collected after adsorption with HP20 macroporous resin contained 20% EA. The recovery of EA from the fruit was 85%. After that, the supernatant was extracted continuously (twice) by column adsorption, and the purified powder with 55% EA content was recovered. This powder was completely dissolved with NaOH and precipitated with HCl, and a precipitated powder with about 85% EA content was collected. Then, the powder was further dissolved in methanol, heated under reflux conditions and the precipitate was collected by centrifugation. The final high-purity powder with 90% EA content was obtained and used for the following evaluation of anticancer activity. Infrared spectra (pure) ν_max_/cm^−1^ 3474, 3155, 1721, 1618, 1582, 1510, 1446, 1398, 1374, 1326, 1260, 1194, 1111, 1038, 922, 875, 813, 757, 687, 634, 580, 535, and 460 (Appendix A). The retention time of EA in the UPLC was 12.38 min. EA retention time in UHPLC was 12.38 min, which was similar to the literature (Appendix A) [41].

### 2.6. Biological Evaluations

#### 2.6.1. Anticancer Activity of EA In Vitro

We further evaluated the anticancer activity of the purified EA on the growth of the human cancer cell lines HeLa, HepG2, MCF-7, and A549 and normal cells (HUVECs) by MTT assay in vitro. The results are shown in Figure 8.

According to Figure 8a, EA-mediated growth inhibition occurred in a dose-dependent manner. The highest inhibition rate was observed in HeLa cells with an IC_50_ of 35 μg/mL, while in MCF-7 cells, the IC_50_ was 73 μg/mL, in HepG2 cells, the IC_50_ was 213 μg/mL, in A549 cells, the IC_50_ was 242 μg/mL, and in HUVECs, the IC_50_ was 258 μg/mL (Figure 8b). The toxicity of EA to the nonmalignant cell line (HUVECs) was also investigated to characterize the selectivity, which is expressed as the selectivity index (SI) (SI = (IC_50_ for nonmalignant HUVECs)/(IC_50_ for the human tumor cell lines)) [42], as illustrated in Figure 8c. An important consideration for future pharmacological applications is the SI. EA exhibits moderate to good cytotoxic activity against human cancer cells, with significantly enhanced selectivity for HeLa, MCF-7, HepG2, and A549 cells (SI value > 1). Li and colleagues further demonstrated the anticancer properties of EA in human cervical cancer cell lines by MTT assays. These authors confirmed that EA reduced the proliferation of human cervical cancer HeLa, SiHa, and C33A cells in a dose- and time-dependent manner, and the inhibitory effect was significantly more pronounced in HeLa cells than in SiHa and C33A cells [30]. In another study, it was previously reported that EA dose-dependently inhibited the growth of HeLa cells [43], which is consistent with our findings.

In order to investigate the apoptosis-inducing ability of EA in HeLa cells, we chose three EA concentrations (50, 25, and 12.5 μg/mL) that were close to the IC_50_ of HeLa cells. Figure 9 is an image of normal HeLa cells in a logarithmic growth phase under a normal light microscope, with clear cell boundaries and moderately pike-shaped morphology. After incubation with different concentrations of EA for different times (24 and 48 h), the morphology and number of HeLa cells were observed under the light microscope. The results showed that the number of cells gradually decreased with the increase in culture concentration and time, and the cells became wrinkled, rounded, and lost their original morphology.

#### 2.6.2. Induction of Apoptosis In Vitro

It is generally recognized that the mechanism of anticancer action of EA involves the regulation of apoptosis. Apoptosis is characterized by numerous morphological changes in the structure of the cell, together with many enzyme-dependent biochemical processes. The result of apoptosis is the removal of cells from the body with minimal damage to surrounding tissues. And the initiation of apoptosis depends on the activation of a series of cysteine-aspartate proteases called caspases [44].

To further investigate the exact mechanism by which EA inhibits the proliferation of HeLa cells, we used flow cytometry to assess the effect of EA on HeLa cell apoptosis. The results indicated that the apoptosis rate of HeLa cells was dose-dependent with EA concentration. The apoptosis rate was highest at about 67% when the EA concentration was 50 μg/mL, 45% when the EA concentration was 25 μg/mL, and 44% when the EA concentration was 12.5 μg/mL. The data showed (Figure 10) that EA significantly inhibited the proliferation of HeLa cells by inducing apoptosis.

#### 2.6.3. Confocal Fluorescence Imaging In Vitro

As mentioned above, EA can inhibit cell proliferation by inducing apoptosis. Therefore, we tried to visualize the induced apoptosis process by confocal fluorescence imaging (Figure 11).

As D’Arcy reported, apoptosis can be distinguished from non-programmed forms of cellular necrosis by visual observation under a microscope and by a variety of molecular biology techniques including flow cytometry and DNA fragmentation assays using Annexin V-FITC staining [44]. After Annexin V-FITC staining, the cytoplasm displays green fluorescence and the PI after staining, the nucleus shows red fluorescence. Apoptotic cells stained only green fluorescence, necrotic cells stained green and red fluorescence, and normal cells did not fluoresce. Almost no green or red fluorescence signal was observed in the negative control, indicating that the cells were in a normal state. After treatment with EA (50 μg/mL) for 48 h at 37 °C, the green fluorescent signal was strong and the red fluorescent signal was weak in the cytoplasm, which indicated that EA had the ability to induce apoptosis.

#### 2.6.4. Apoptosis Pathway Assay

In order to further determine the pathways by which EA inhibits the growth of HeLa cells, we studied the changes in the mRNA levels of genes associated with the apoptotic pathway. Changes in apoptosis-related pathway genes are shown in Figure 12. The results showed that *caspase*, *PTEN*, *TSC*, and *mToR* expression were up-regulated and *AKT*, *PDK1* expression were down-regulated, which showed a dose-dependent relationship with EA. *E6/E7* are two sequences encoded by HPV18 whose open reading frames are directly involved in cervical cancer cell growth and development. We further evaluated the *E6/E7* gene. The mRNA levels of *E6/E7* were determined to decrease gradually with the increase in EA incubation time and concentration. The phosphatase and tensin homolog (PTEN), which is deleted on chromosome 10, is a tumor suppressor that negatively regulates the AKT/PI3K with negative regulatory effects, and the AKT/PI3K signaling pathway is associated with tumorigenesis and apoptosis [45,46]. The mammalian target of rapamycin (mTOR) is involved with malignancy-associated genes, which can promote cancer cell proliferation and inhibit apoptosis [47]. TSC upstream of the mTOR signaling pathway, and the TSC1/TSC2 complex can inhibit mTOR activity [48]. The above results tentatively suggest that EA may inhibit cell growth through the above-related pathway.

## 3. Discussion

It has been shown that daily consumption of phenolic-rich foods has a preventive effect on some diseases [49,50], so it has a positive effect on the breeding of plants with high phenolic acid content. In this paper, we initially determined the trend of EA content during the growth and development process of Hull fruits; the results showed that Hull has high EA content. The highest EA content was 5.76 mg/g FW in the S1 period and the lowest EA content was in the S5 period. Because metabolites during fruit ripening depend on the expression of relevant genes and the action of biological enzymes [51], metabolomics and transcriptomics are of great significance for the study of metabolite accumulation. Therefore, elucidating the molecular mechanism of EA accumulation in blackberry fruits will have a positive effect on the targeted selection of superior varieties, and it will be beneficial to develop effective methods to improve fruit quality in response to market demand.

As verified by in vitro cell assay, the results indicated that EA could regulate the expression of *E6/E7* genes through PTEN/AKT/mTOR/PI3K-related pathways and inhibit the growth and development of HeLa cells. In a few words, EA inhibits the growth of HeLa cells by inducing apoptosis. Apoptosis is a form of cell suicide in a physiological mode, resulting in controlled cell death. It plays a central role in cancer and can provide important information for studying behaviors related to development and homeostasis regulation. Consistent with Li’s findings [30], E6/E7 is directly involved in the growth and development of cervical cancer cells, and EA has an inhibitory effect on its expression, so EA shows an inhibitory effect on HeLa. As per our previous report, polyphenol extract may interact with DNA in an intercalation mode to change or destroy DNA and cause apoptosis, and inhibit cell proliferation. When DNA damage is introduced into cells from exogenous or endogenous sources there is an increase in the amount of intracellular reactive oxygen species (ROS) that may be related to apoptosis [42]. EA is a kind of natural polyphenol, the further possible mechanism may be that EA can induce apoptosis by interacting with DNA in an intercalation mode, changing or destroying DNA. However, EA cannot be exploited for in vivo therapeutic applications in the current situation because of its poor water solubility and accordingly low bioavailability. Thus, further work to improve its water solubility and bioavailability will be needed in the future. Despite the fact that blackberry is one of the natural sources of EA, an edible berry rich in several phenolic acids, the digestive and absorptive transformation of EA after ingestion into the human body is necessary. As Lei [52] reported that EA in pomegranate leaf is rapidly absorbed and distributed as well as eliminated in rats. Zhou and co-workers determined the pharmacokinetics of oral EA in rats with rapid distribution and time to peak. The blood concentration peaked at 0.5 h with C_max_ = 7.29 μg/mL, and the drug concentration decreased to half of the original after 57 min of administration [53]. Therefore, increasing the drug concentration, prolonging the retention time of EA, and improving the bioavailability are of great significance in the research and development as well as utilization of botanical drugs.

## 4. Materials and Methods

### 4.1. Materials and Chemicals

Hull blackberry fruits were collected from the Baima Science Research Base of the Institute of Botany, Jiangsu Province, and Chinese Academy of Sciences (Nanjing, China), and the disease-free fruits were stored at −30 °C. EA standards (content > 96%, were used as control analysis; content > 98%, standard products). The total antioxidant capacity assay kit, DPPH free radical scavenging capacity assay kit, plant soluble sugar content test kit, Vitamin E assay kit, Vitamin C assay kit, and fructose assay kit (Nanjing Jiancheng Bioengineering Institute, Nanjing, China) were used. The macroporous resins HPD100, HPD300, HPD600, AB-8, and HP20 (Sobolai, Beijing, China) and acetonitrile, trichloroacetic acid, potassium bromide (KBr), Dulbecco’s modified Eagle’s medium (DMEM), fetal bovine serum (FBS), phosphate-buffered saline (PBS), 3-(4,5-dimethyl-2-thiazolyl)-2,5-diphenyl-2-H-tetrazolium bromide (MTT), dimethyl sulfoxide (DMSO), penicillin/streptomycin, trypsin, the RNA extraction kit (Keybionet, Nanjing, China), and Annexin V-FITC/PI (Beyotime, Shanghai, China) were purchased commercially. HeLa, HepG2, MCF-7, and A549 cells and HUVECs were kindly provided by Cell Bank, Chinese Academy of Sciences.

### 4.2. Measurement Method of EA

#### 4.2.1. UV Spectrophotometric Method

The EA content was rapidly determined by UV spectrophotometry. EA is a phenolic acid that reacts with bases, and this complex has a maximum absorption peak of 357 nm [54]. According to this method the gradient concentration EA absorbance values were determined and the standard curve was plotted as in Figure 13. The standard curve gave the equation of y = 0.0531x + 0.0042 after fitting (R^2^ = 0.9997).

#### 4.2.2. UHPLC Analysis

An Agilent 1260 Infinity II ultra-high performance liquid chromatography system was used for separation by Phenomenex Gemini 5u C18 column (250 mm × 4.60 mm, 5 μm). The mobile phase water(A)/acetonitrile(B) (A:B = 85:15) acidified to 0.05% (*w*/*v*) with trichloroacetic acid. The flow rate was 1 mL/min; the column temperature was 30 °C; the wavelength of 254 nm; and the injection volume of 10 μL [55].

### 4.3. Measurement of Fruit Quality Indexes

Ultrasound-assisted solvent extraction method was used to extract the anthocyanin in the fruits, and anthocyanin content was determined at 510 nm by UV spectrophotometry [56]. Determination of the total phenol content of the fruit was carried out by Floin–Ciocalteu method [57], absorbance value was measured at 750 nm, and the total phenol content in the sample was calculated by the standard curve y = 0.128x + 0.00045 (R^2^ = 0.999). The content of the remaining substances was determined according to the method of the kit instructions.

### 4.4. Single-Factor Experiments of EA Extraction from Blackberries

For each stage, 50 g of fruit was homogenized and prepared. Extraction conditions were as follows: the temperature of ultrasonic extraction was 50 °C, the time of ultrasonic extraction was 20 min, the solid–liquid ratio was 1:10, and the solvent was anhydrous ethanol. After the extraction, the extract was centrifuged at 7000 rpm for 5 min, and then 1 mL of the clarified supernatant was added with 4 mL of NaOH (0.1 mol/L), absorbance value was measured at 357 nm, so as to calculate the concentration of EA.

An analysis of four extraction process factors was performed to determine the effect of each factor on extraction efficiency in order to optimize the extraction process: solid–liquid ratio (1:5, 1:10, 1:15, 1:20, 1:25, 1:30, 1:35, and 1:40 (g:mL)), ethanol concentration (100%, 90%, 80%, 70%, 60%, 50%, 40%, 30%, 20%, and 10%), ultrasonic time (10 min, 20 min, 30 min, 40 min, and 50 min) and extraction temperature (50 °C, 60 °C, 65 °C, 70 °C, 75 °C, 80 °C, 85 °C).

### 4.5. Response Surface Factor Test

From the results of single-factor experiments, the Box–Behnken design in Design-Expert v8.0 software was used to optimize the extraction process with ethanol concentration (A), solid–liquid ratio (B), extraction time (C) and extraction temperature (D) as the independent variables, and amount of EA extracted as the dependent variable. Thus, −1, 0 and 1 denote the three levels of the independent variables. The experimental factors and levels were designed as shown in Table 3.

### 4.6. EA Purification Experiment

After pretreatment, five macroporous resins of different polarities were selected, which include polar HPD600, weakly polar AB-8 and HPD300, and non-polar HP20 and HPD100. Next, 100 mL triangular flasks were filled with 3 g of the resins and 50 mL of the EA supernatant, and shaken for 5 min, 10 min, 15 min, 30 min, 60 min, 120 min, 180 min, 240 min, 300 min, and 360 min, at 150 rpm at room temperature. The EA content was determined from these data and the adsorption curves were plotted. We rinsed the EA-adsorbed resin with water to remove excess contaminants and liquid, and then added 50 mL of 60% ethanol and took samples at the same points to plot the resolution curves in the adsorption experiments. The most effective resin was selected for subsequent purification experiments.

The extracted supernatant was purified by column chromatography and eluted with 60% ethanol, and the primary purified EA powder was obtained after evaporation of ethanol and freeze-drying. The primary purified EA powder was mixed with 1 mol/L NaOH until it was completely dissolved, the pH was adjusted to 12~14, the pH was then adjusted to 2~4 by adding HCl, and EA was precipitated overnight at 4 °C. The EA precipitate was collected by centrifugation after secondary purification, and the secondary purified product was added to a methanol solution at a ratio of 1:100. After heating to reflux at 80 °C for 2 h, the solution was recrystallized overnight at 4 °C and collected. Finally, the relative content of EA was determined.

### 4.7. MTT Assay In Vitro

Human cervical cancer cells (HeLa), human non-small-cell lung cancer cells (A549), human breast cancer cells (MCF-7), and human liver cancer cells (HepG2) were cultured in vitro under 5% CO_2_ at 37 °C. Cancer cells were added to 96-well plates at a density of 1 × 10^4^ cells per well. After 12 h of incubation, the medium was removed and the cells were incubated with different EA concentrations (each concentration was repeated three times) for 48 h. Then, the medium was removed and a new medium containing MTT (1 mg/mL) was added and incubated for 4 h. The absorbance of each well was measured at 595 nm. Cell viability values were determined (at least three times) according to the following formula: cell viability (%) = absorbance of the experimental group/absorbance of the blank control group × 100% [54,57].

### 4.8. Induction of Apoptosis In Vitro

HeLa cells (1 × 10^6^) were cultured in 35 mm dishes and incubated at 37 °C for 24 h. After adding EA at concentrations of 12.5, 25, and 50 μg/mL, the cells were incubated for 48 h (each concentration was repeated three times). DMSO (0.1%) was used as a control. The treated cells were washed, trypsinized (without EDTA (ethylene diamine tetraacetic acid), and centrifuged. Next, the cells were collected and resuspended in 500 μL of buffer solution loaded with 5 μL of Annexin V-FITC and 5 μL propidium iodide (PI), the cells were incubated for 5–15 min in the dark. Flow cytometry analysis was conducted with a single 488 nm argon laser and 80,000 events with a BD Accuri C6 flow cytometer and software (Becton, Dickinson and Company, Franklin Lakes, NJ, USA) [54].

### 4.9. qRT–PCR Detection

After treating the cells with the same procedure as in vitro apoptosis assessment, all the cells were collected to extract all the RNA according to the kit method and reverse transcribed to cDNA. cDNA was diluted to 400 ng/μL. The fluorescence quantification reaction (qPCR) system had a total volume of 15 μL (1 μL of cDNA, 0.6 μL of primers, 5.3 μL of ddH_2_O, and 7.5 μL of mix). The primer sequences are detailed in Table 4.

### 4.10. Data Analysis

The analytical data were statistically analyzed using SPSS Statistics 16.0 ((IBM, Chicago, IL, USA) and one-way ANOVA at a level of reliability of 0.05%. Origin 2019b (OriginLab Crop., Northampton, MA, USA) was used to analyze the IC_50_ values with linear fitting. All experiments were repeated three times.

## 5. Conclusions

As one of the earliest introduced blackberry varieties in China, the Hull variety is currently planted over a large area in Jiangsu Province [58]. Hull is an extremely productive variety, has a high resistance to stress, and is the parent of many bred blackberry varieties [59,60]. Research has shown that the aging process and the occurrence of some diseases are oxidative processes, and the daily consumption of phenolic acid-rich substances has a positive effect on the occurrence and alleviation of some diseases [61,62]. Blackberries are rich in a variety of antioxidant-active substances such as phenols, flavonoids, and vitamins [63]. Blackberry fresh fruits are not easy to store, so the trend of blackberry active substances as well as their extraction and purification have certain far-reaching significance for the development and utilization of blackberries. In this research, in order to further develop and utilize blackberry fruits in depth, the fruits were divided into different developmental periods according to their color. The results showed that the EA content decreased gradually with the growth of blackberry fruits. The total antioxidant capacity, DPPH radical scavenging capacity, and total phenols in the fruit showed similar trends to the EA content, which was the highest in the S1 stage. The anthocyanin and saccharides accumulated gradually with the growth and development of the fruits, and the highest content was found at the S5 stage. The extraction process of EA was optimized by single-factor and response surface test, and the high-purity blackberry EA powder was purified several times. The results showed that EA had an anticancer effect and inhibited the growth of HeLa cells by inducing apoptosis.

## Figures and Tables

**Figure 1 ijms-24-15228-f001:**
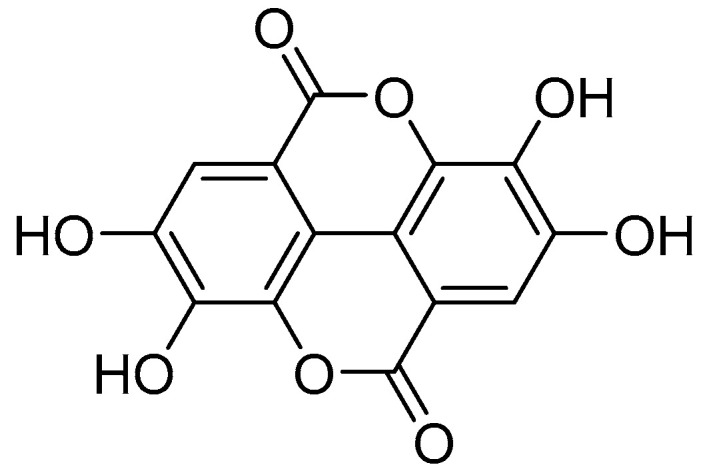
Chemical structure of ellagic acid.

**Figure 2 ijms-24-15228-f002:**
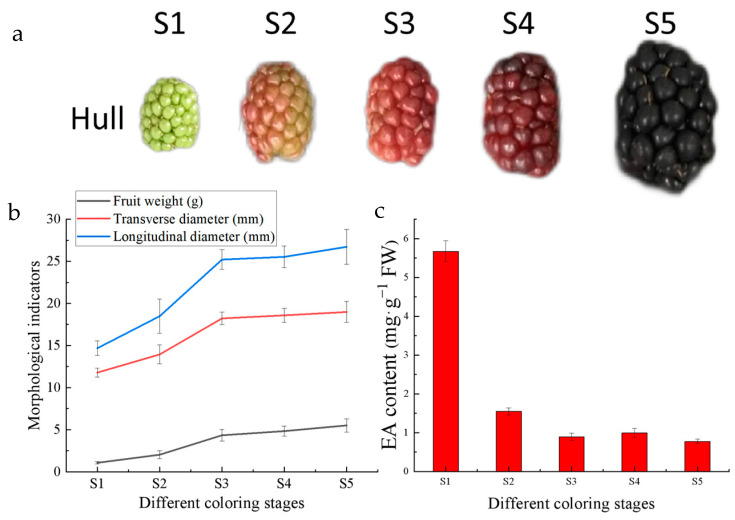
Hull fruit indicators: (**a**) Hull fruit appearance at each stage; (**b**) morphological indicators of Hull fruits at different coloring stages; (**c**) EA content in Hull fruits at different coloring stages.

**Figure 3 ijms-24-15228-f003:**
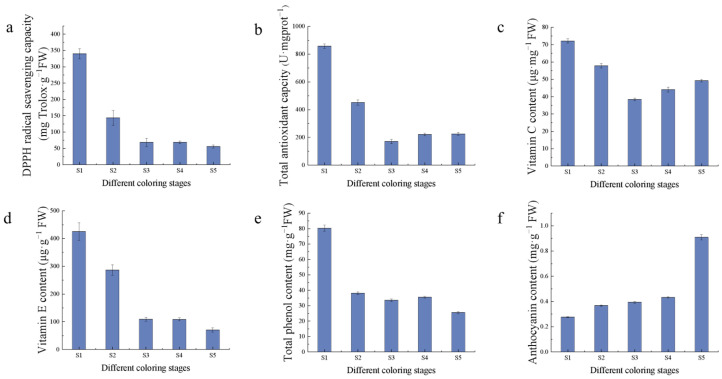
Antioxidant indicators: (**a**) DPPH radical scavenging capacity; (**b**) total antioxidant capacity; (**c**) vitamin C content; (**d**) vitamin E content; (**e**) total phenol content; (**f**) anthocyanin content.

**Figure 4 ijms-24-15228-f004:**
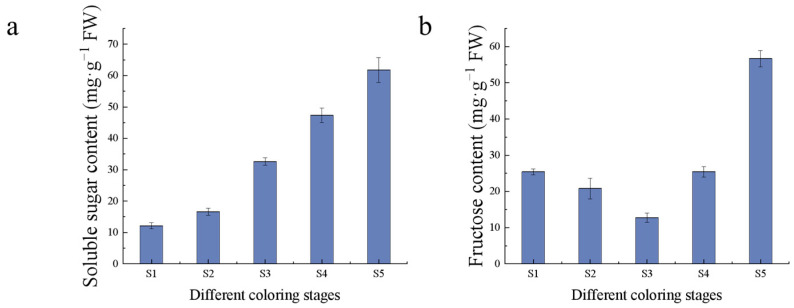
Sugars content. (**a**) Soluble sugar content; (**b**) fructose content.

**Figure 5 ijms-24-15228-f005:**
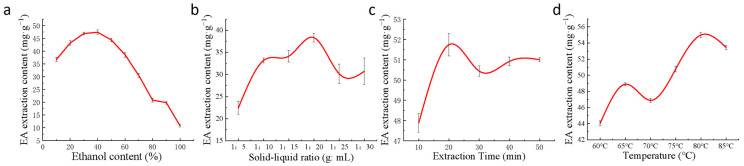
The effect of each single factor on the extraction efficiency of EA. (**a**) Ethanol concentration; (**b**) solid–liquid ratio; (**c**) ultrasonication time; (**d**) ultrasonication temperature.

**Figure 6 ijms-24-15228-f006:**
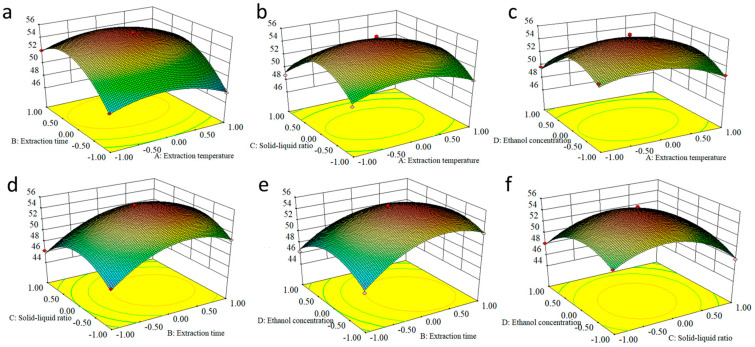
Response surface maps of the corresponding quadratic regression equations. (**a**) AB; (**b**) AC; (**c**) AD; (**d**) BC; (**e**) BD; (**f**) CD.

**Figure 7 ijms-24-15228-f007:**
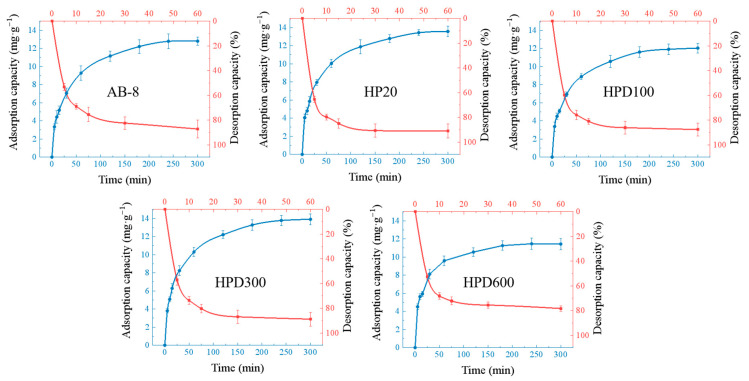
Adsorption and resolution curves with the five macroporous resins. The blue lines are the adsorption curves, and the red lines are the resolution curves.

**Figure 8 ijms-24-15228-f008:**
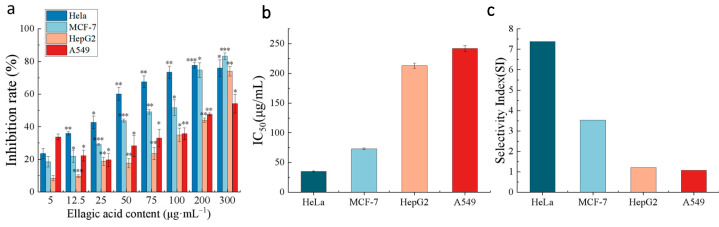
The anticancer activity of the purified EA extract. (**a**) The rates of inhibition of four types of cancer cells induced by EA; (**b**) IC_50_ values of the purified EA extract against four types of cancer cells; (**c**) selectivity indices (SIs) of the purified EA extract. *** *p* < 0.001; ** *p* < 0.01; * *p* < 0.05.

**Figure 9 ijms-24-15228-f009:**
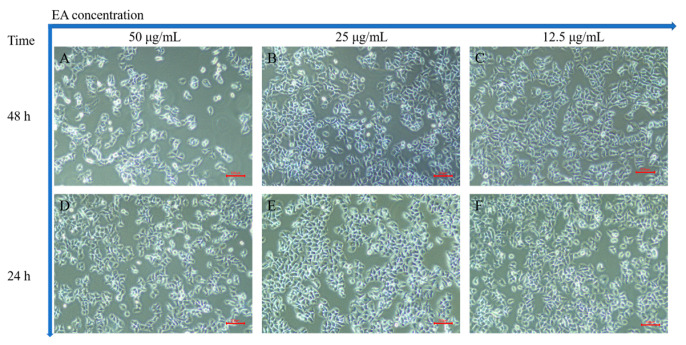
HeLa cells incubated with EA. (**A**–**C**) are images taken under a light microscope after incubation with 50, 25, and 12.5 μg/mL EA for 48 h, respectively; (**D**–**F**) for 24 h, respectively. Scale bar = 500 μm.

**Figure 10 ijms-24-15228-f010:**
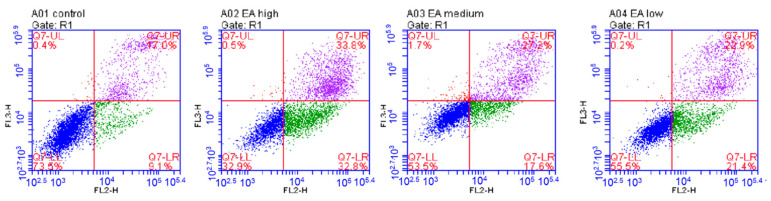
Apoptosis induced by EA analyzed by flow cytometry. HeLa cells incubated with DMSO were used as controls, while experimental cells were treated with 50, 25, and 12.5 μg/mL EA.

**Figure 11 ijms-24-15228-f011:**
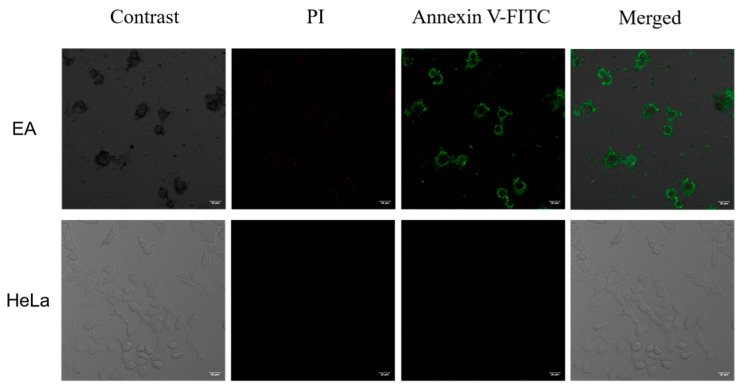
Confocal fluorescence microscopy images of HeLa cells incubated with 50 µg/mL EA for 48 h at 37 °C. Scale bar = 20 μm.

**Figure 12 ijms-24-15228-f012:**
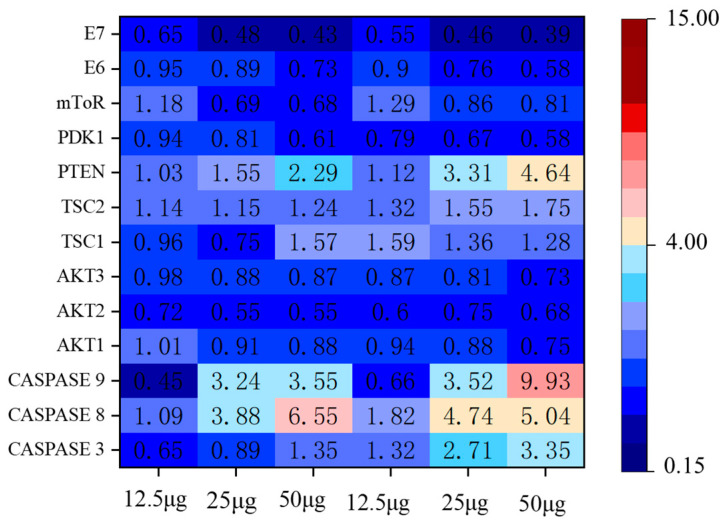
Heatmap of the inhibition of HeLa cell growth and apoptosis pathway-related genes after incubation with 12.5, 25, and 50 μg/mL EA for 24 h and 48 h.

**Figure 13 ijms-24-15228-f013:**
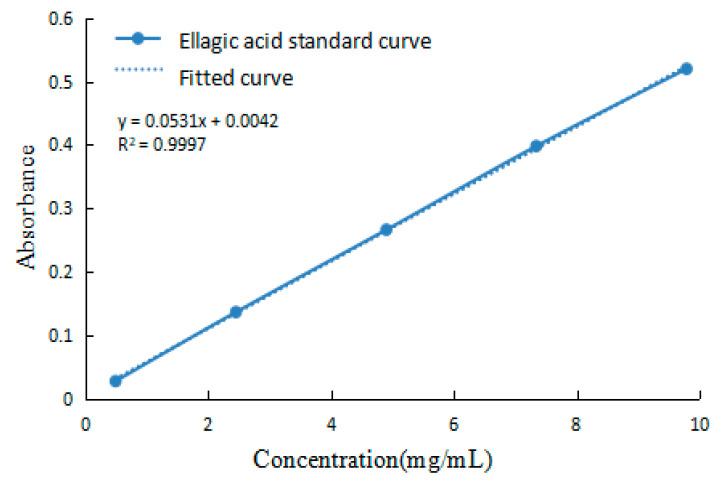
EA standard curve.

**Table 1 ijms-24-15228-t001:** Response surface design experimental setup and results.

Number	A (°C)	B (min)	C (mL/g)	D/%	Ellagic Acid Content (mg/g)
1	1	0	1	0	48.28
2	0	0	0	0	54.84
3	1	1	0	0	51.24
4	0	−1	1	0	46.26
5	0	0	−1	1	48.12
6	−1	0	−1	0	50.21
7	0	1	1	0	49.06
8	−1	0	1	0	48.73
9	0	−1	−1	0	47.22
10	0	1	0	1	48.82
11	0	1	−1	0	50.83
12	1	0	0	−1	50.75
13	0	0	0	0	54.86
14	−1	−1	0	0	48.62
15	1	0	0	1	49.51
16	0	−1	0	1	46.36
17	0	0	1	1	47.11
18	0	0	1	−1	47.95
19	−1	0	0	1	49.70
20	1	0	−1	0	50.10
21	−1	1	0	0	52.04
22	0	0	0	0	55.55
23	0	1	0	−1	52.18
24	−1	0	0	−1	52.50
25	1	−1	0	0	47.52
26	0	0	0	0	55.72
27	0	−1	0	−1	46.87
28	0	0	−1	−1	50.95
29	0	0	0	0	54.71

**Table 2 ijms-24-15228-t002:** Analysis of variance (ANOVA) of the fitted quadratic polynomial model.

Source	Sum of Squares	DF	Mean Square	F Value	*p* Value
Model	223.41	14	15.96	80.66	<0.0001
A	1.61	1	1.61	8.16	0.0127
B	37.85	1	37.85	191.3	<0.0001
C	8.42	1	8.42	42.55	<0.0001
D	11.16	1	11.16	56.42	<0.0001
AB	0.023	1	0.023	0.12	0.7385
AC	0.029	1	0.029	0.15	0.708
AD	0.62	1	0.62	3.12	0.099
BC	0.17	1	0.17	0.85	0.3734
BD	2.04	1	2.04	10.34	0.0062
CD	0.99	1	0.99	5	0.0422
A^2^	22.71	1	22.71	114.81	<0.0001
B^2^	74.74	1	74.74	377.81	<0.0001
C^2^	87.38	1	87.38	441.69	<0.0001
D^2^	55.37	1	55.37	279.86	<0.0001
Residual	2.77	14	0.2		
Lack of fit	1.91	10	0.19	0.88	0.6044
Pure error	0.86	4	0.22		
Cor total	226.18	28			
R-squared	0.9878				
Pred R-squared	0.9455				
Adj R-squared	0.9755				

Note: differences were considered significant at the level of *p* < 0.05.

**Table 3 ijms-24-15228-t003:** Single-factor level experimental design.

Factors	Code	Coding Level		
		−1	0	1
Ethanol concentration	A	30%	40%	50%
Solid–liquid ratio	B	1:15	1:20	1:25
Extraction time	C	10 min	20 min	30 min
Extraction temperature	D	75 °C	80 °C	85 °C

**Table 4 ijms-24-15228-t004:** Primer sequences for quantitative fluorescence analysis.

Gene Name	Forward (5′-3′)	Reverse (5′-3′)	Amplification Length (bp)
*Caspase 3*	ACCAGTGGAGGCCGACTTCT	GCATGGCACAAAGCGACTGG	107
*Caspase 8*	ACCGAAACCCTGCAGAGGGA	CATCGCCTCGAGGACATCGC	78
*Caspase 9*	GATGCCCTGTGTCGGTCGAG	GTGGAGGCCACCTCAAACCC	139
*AKT1*	GCGGCACACCTGAGTACCTG	CAGGCGACCGCACATCATCT	113
*AKT2*	CAGAACACCAGGCACCCGTT	GCCCGCTCCTCTGTGAAGAC	143
*AKT3*	TGTCGAGAGAGCGGGTGTTC	TGGTGGCTGCATCTGTGATCC	198
*PTEN*	TCCCAGTCAGAGGCGCTATGT	CCGTCGTGTGGGTCCTGAAT	199
*TSC1*	AAGCTTGGGCCTGACACACC	CTGTCTCCCGCAGGGCTTTC	86
*TSC2*	ATCTGCAGCGTGGAGATGCC	GTGTACGGCAGGGAGATGGC	200
*mTOR*	GCCTTTCCTGCGCAAGATGC	GCGGGCACTCTGCTCTTTGA	85
*PI3K*	AGGAGATCGCTCTGGCCTCA	TGGCTCGGTCCAGGTCATCC	161
*E7*	GGACGGGCCAGATGGACAAG	GGGTTCGTACGTCGGTTGCT	122
*E6*	TGTGTCAGGCGTTGGAGACAT	ACCTCAGATCGCTGCAAAGT	82

## Data Availability

The datasets generated during and/or analyzed during the current study are available from the corresponding author upon reasonable request.

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
