# Peer review of "Ellagic Acid from Hull Blackberries: Extraction, Purification, and Potential Anticancer Activity"

_ijms, 2023, doi:10.3390/ijms242015228_

Round 1

Reviewer 1 Report

The authors examined the extraction and purification of ellagic acid (EA) from Hull blackberries and its possible anti-tumor activity. They showed that EA showed a strong inhibitory effect on cell proliferation in HeLa cells, suggesting that the mechanism of the anti-tumor activity of EA may be due to its apoptotic action via the PI3K/PTEN/AKT/mTOR pathway. I believe that this manuscript is consistent with the main purpose of this special issue of IJMS, and that the experimental methods, results, and discussion are systematic and detailed. However, the following minor revisions are needed.

Please describe in the discussion section why EA showed strong anti-tumor activity in HeLa cells, a possibility that the authors consider.

A limitation of this manuscript is that it does not include data on the anti-tumor effects of EA on in vivo animal models. Free radical scavengers such as vitamin E and N-acetylcysteine have been reported to promote prostate cancer growth in experimental animal studies or clinical trials. Please describe in your discussion the challenges and prospects for future in vivo anti-tumor experiments and clinical trials of EA.

Reviewer 2 Report

The manuscript submitted for review is well thought out and prepared. It contains a lot of valuable data from correctly conducted experimental studies.

I suggest considering the following points when preparing the revision:
1) I feel it would be useful to show the structure of the EA.
2) The following sentence in the Introduction is probably not needed: Cancer is a complex multistage process that begins with initiation of cancer cells including DNA damage, accumulation of mutations, promotion of cell proliferation and tumor expansion, and ends with malignancy and metastasis [24]. (l. 53-56)
3) The word "cancer" is repeated three times in this sentence: Cervical cancer is a common cancer and the fourth leading cause of cancer death in women [30]. (l. 58 and 59)
4) l. 86. It is: Fig. 2b. Should be: Fig. 1b.
5) At the end of this sentence, I would give the numerical value as for the S1 stage: Among the five stages, the highest EA content was found at the S1 stage at 5.67 mg/g (Fig 1c), while the lowest EA content was found at the S5 stage. (l. 87-89)
6) Figs. 4 and 6. In my opinion, the trend line was poorly chosen. The broken line used would indicate that there is a linear relationship between the experimental points, which changes in nature between points. A "smooth" trend line should have been selected.
7) In my opinion, numerical values expressed in percentages should be rounded to whole numbers if possible. Because percentages are for rather only illustrative purposes. Example in the following sentence: The adsorption rates were about 64.56% and 62.94%, and the amounts of EA adsorbed on the resins were 13.77 mg/g and 13.43 mg/g, respectively. (l. 204-206)
8) Please show the FTIR spectrum in the Supporting Materials. What does "Vmax" (l. 230) mean for the infrared spectrum? If it is supposed to be a wave number, a different symbol is used. The wave numbers mentioned (l. 230 and 231) do not have a unit.
9) Please also show the UPLC chromatogram in the Supporting Materials. (l. 231-233)
